# Sport Activity Load and Skeletomuscular Robustness in Elite Youth Athletes

**DOI:** 10.3390/ijerph19095083

**Published:** 2022-04-21

**Authors:** Irina Kalabiska, Annamaria Zsakai, Dorina Annar, Robert M. Malina, Tamas Szabo

**Affiliations:** 1Center for Sport Physiology, University of Physical Education, Research, Alkotas u. 44, 1123 Budapest, Hungary; kalabiskai@gmail.com (I.K.); szabo.tamas@mksz.hu (T.S.); 2Department of Biological Anthropology, Eotvos Lorand University, Pazmany p. s. 1/c., 1117 Budapest, Hungary; annar.dorina@gmail.com; 3Department of Kinesiology and Health Education, University of Texas at Austin, Austin, TX 78712, USA; rmalina@1skyconnect.net; 4Sport Sciences and Diagnostic Research Centre, Hungarian Handball Federation, Konyves K. Krt. 76, 1087 Budapest, Hungary

**Keywords:** bone mineral, skeletomuscular robusticity, elite athletes, DEXA

## Abstract

In an earlier report, bone mineral reference values for young athletes were developed. This study addressed variations in bone mineral parameters of young athletes participating in sports with different mechanical loads. The bone mineral status of 1793 male and female athletes, 11 to 20 years of age, in several sports was measured with DEXA. Specific bone mineral parameters were converted to z-scores relative to age- and sex-specific reference values specified by the DEXA software. Z-score profiles and principal components analyses were used to identify body structural components in the young athletes and to evaluate the associations between the identified component and type of sport defined by mechanical load. A unique skeletomuscular robusticity of male wrestlers, pentathletes, and cyclists was noted: wrestlers had significantly more developed skeletomuscular robusticity and bone mineral density compared to the age-group average among elite athletes, while pentathletes and cyclists had lower bone mineral parameters than the age-group references among elite athletes. Among female athletes, bone mineral parameters of both the trunk and extremities of rhythmic gymnasts and pentathletes were significantly lower compared to the age-group means for elite athletes. The bone mineral development of elite young athletes varies with the impact forces associated with their respective sports. The skeletal development of cyclists, pentathletes, and rhythmic gymnasts should be monitored regularly as their bone development lags behind that of their athlete peers and the reference for the general population.

## 1. Introduction

Athletes are not a homogenous population as both intrinsic (genetic and biological) and extrinsic (environmental, nutritional, and training) factors affect performance [1]. Both regular training for sport [2] and regular participation in physical activity [3] generally have a beneficial effect on mineral accrual and, correspondingly, bone mass and bone mineral density (BMD). The skeleton in childhood and adolescence is sensitive to the mechanical stimulation elicited by physical activity so that regular physical activity during childhood and adolescence can optimize skeletal health that persists through adulthood. Factors other than mechanical stimulation also influence bone development and include genotype and adequate levels of vitamin D and calcium [4,5,6]. Although BMD among athletes systematically training in different sports has received considerable attention [7,8,9], variations in total body BMD and BMD of body segments among athletes training in different sports merits attention.

Longitudinal training studies indicate that strength training and high-impact endurance training (anaerobic exercises) are associated with an increase in BMD [10]. On the other hand, negative effects of training (i.e., reduced BMD) in distance runners have also been reported [11,12]. Evidence for beneficial effects of non-weight-bearing activities (water sports) on bone mineral accrual and maintenance is somewhat controversial [13,14], and it has been questioned whether the effects are similar to those observed in athletes participating in weight-bearing sports [15,16]. There is also some indication that weight-bearing aerobic exercise may be more beneficial for bone health than non-weight-bearing activities [17]. Nevertheless, the most effective training protocol for attaining and maintaining a high bone mass and BMD has not been firmly established. Maintaining high bone mass and strength has an important role in the prevention of stress fractures and other skeletal injuries.

The growth status of children and adolescents, including youth athletes, is routinely evaluated relative to an established growth reference appropriate for a geographical region and, perhaps for the time of study, a given population variability and secular trends in growth [18,19]. Allowing for the selectivity of many sports (e.g., size and physique) and systematic effects of training on body composition and specific components of body composition, it has been suggested that reference values for athletes in specific sports may be more appropriate [20]. Although systematic training for sport influences the body composition of youth athletes, training does not influence linear growth nor growth in stature [21].

Reference values for several bone mineral parameters among youth athletes that are 10–20 years old have been developed [20] in the context of the need for reference data for national and athlete-specific samples. The initial results, though of interest, also suggested a potential need for bone mineral reference data for youth athletes training in specific sports. In this context, the purpose of the present study was to compare variations in bone mineral parameters of young athletes participating in sports with different mechanical loads.

The asymmetries of skeletomuscular development between the upper and lower body segments (e.g., rowers versus cyclers) or between the dominant and non-dominant arms (e.g., tennis players versus swimmers) associated with training load are reasonably well-established. Skeletomuscular development is usually significantly asymmetric in favor of the dominant side or region of the body, although an increased level of physical activity may help to prevent incorrect body posture, while asymmetric training loads on skeletal muscles may also enhance incorrect posture [22,23,24,25]. It is not clear, however, if this developmental asymmetry is also manifest in bone mineral parameters of body segments with different training loads. Thus, a secondary objective of this study was to compare bone mineral parameters of the upper and lower extremities and of the trunk among young athletes participating in different types of sports.

## 2. Materials and Methods

### 2.1. Study Design

This project was approved by the Research Ethics Committee of the University of Physical Education in Budapest, Hungary (ID of approval: TE-KEB/No42/2019). The Research Center for Sport Physiology (University of Physical Education, Hungary) has a cooperation agreement with numerous sports federations, associations, and clubs that focus on a variety of sports. The governing bodies of the respective sports organizations also approved the ethical codes established by Research Ethics Committee of the university. The research was carried out following the rules of the Declaration of Helsinki of 1975 (https://www.wma.net/what-we-do/medical-ethics/declaration-of-helsinki/, accessed on 21 December 2021) as revised in 2013. Parents of athletes <18 years old and the athletes were informed of the details of the project; both provided written informed consent. Details of the project were also provided for older athletes who also provided informed written consent.

### 2.2. Subjects

Subjects included a cross-sectional sample of 1734 athletes, 1299 males (11–20 years old) and 435 females (13–20 years old), who voluntarily agreed to participate in the study (Table 1 and Table 2) [17]. The athletes represented several sports academies—primarily basketball, football, and handball, with smaller numbers for several individual sports such as pentathlon, rhythmic gymnastics, kayak, canoe, rowing, and wrestling. The athletes trained regularly, several times per day in many instances, and also regularly participated in competitions and tournaments on weekends. The athletes, as a group, had been training in their respective sports since 6–7 years of age and were considered elite.

### 2.3. Data Collection

The cross-sectional research was conducted between September 2015 and February 2020. Whole-body bone mineral density (BMD), bone mineral content (BMC), and body mass components (fat mass, muscle mass) were measured with a GE Lunar Prodigy dual-energy X-ray scanner (GE Medical Systems, Madison, WI, USA).

Subjects were asked to avoid eating and drinking for at least 60 min prior to examination and to follow their habitual training regime during the week of the examination. All examinations took place between 9:00 and 12:00 in the morning.

Several athletes were excluded from the study if they:Had severe degenerative lesions or fractures/deformations in the measurement area;Were unable to reach the correct position and/or remain immobile during measurement;Had a very high or low body mass index (i.e., a BMI that could adversely affect the accuracy of measurement process);Were exposed to an enhanced X-ray/CT scan several days prior to the study;Were pregnant.

### 2.4. Statistical Analyses

The body mass components and bone mineral densities were expressed relative to height (m) to reduce the influence of body size on the specific parameters. The structural and bone mineral parameters of each athlete were converted to z-scores relative to age- and sex-specific mean and standard deviation values estimated for the total sample (z = (individual value–age-group mean) / age-group standard deviation). Bone mineral densities of the upper extremities, lower extremities, and trunk were also expressed as a percentage of total BMD to address asymmetries in BMD by type of sport [26].

Wilcoxon signed-rank test was used to compare the z-scores of components of body mass and of bone parameters in the athletes by sport (an alpha of 0.05 was used as the cut-off for significance in all analyses). Linear regression analysis was used to evaluate the relationship between the L1–L4 BMD z-scores estimated by the GE Prodigy Lunar reference and the youth athlete reference series.

Principal components analysis (PCA) was used to reduce the body parameters to a smaller set of components that accounted for most of the variance in the variables considered. Sex-specific PCAs were initially conducted; as the analyses showed similar components and loadings, the analysis was also conducted for the total sample. The original variables were log-transformed to approximate the normality assumption of PCA. Based on eigenvalues >1.0, loadings >0.90 were used to identify the variables characteristic of the respective components. Cronbach’s alpha was used to estimate the reliability of the analysis. Principal component scores were also calculated for each athlete.

## 3. Results

The z-score profiles of body mass components and bone parameters indicated greater skeletal and muscular robustness of wrestlers compared with male participants in other sports, especially pentathletes, whose skeletomuscular robustness was the least developed among the sports represented in the sample. The skeletomuscular robustness of cyclists was also less developed compared to the age-group reference, but their muscular development was similar to the age-group mean value for males (Figure 1, Table 3). Muscle mass was also highly developed among rowers, kayaker–canoeists, and handball players.

The skeletomuscular development of female basketball and handball players was greater than the age-group average, although the bone mineral density of basketball players was slightly less compared to handball players (Figure 2, Table 3). The bodily structure of rhythmic gymnasts and pentathletes, on the other hand, differed significantly from athletes in the two team sports; their average skeletomuscular development and fatness were also below the age-group means (Figure 2).

Results of the principal components analysis are summarized in Table 4. Two components were indicated: PC 1 described skeletomuscular robustness identified by body mass, total body BMC, and muscle mass, while PC 2 described body composition in the context of BMD (positive) and body fatness (negative). PC 1 accounted for more than 60% of the total variance, while PC 2 explained an additional 20%. Overall, the results suggested that skeletomuscular robustness is the main source of variance in the overall body structure and bone development of youth athletes.

The distribution of PC scores for each component (PC 1 x-axis, PC 2 y-axis) by sport is illustrated in Figure 3 for both males and females. The skeletomuscular robustness of wrestlers (upper right of the plot) stood out relative to male athletes in other sports. The position of the male cyclists and pentathletes (lower left) highlighted their lower level of overall skeletomuscular development and lower BMD compared to athletes in the other sports. Nevertheless, the overlap of the distribution of PC scores should be noted as it highlights the variability among individual athletes in the respective sports.

Among female athletes, the position of rhythmic gymnasts and pentathletes relative to handball and basketball players stood out (Figure 3). In contrast to the overlap among male athletes in different sports, the overlap of rhythmic gymnasts and pentathletes compared to handball and basketball players was minimal.

Median z-scores for BMC and for muscle mass of the upper and lower extremities and of the trunk are illustrated in Figure 4, while results of the comparisons of athletes by sport are summarized in Table 5. Among male athletes, median z-scores for BMC in the upper extremities and the trunk of wrestlers and handball players were above the age-group average, while the BMC of the lower extremities was above the age-group average in both handball and basketball players (Figure 4, upper part). In contrast, the BMC of the lower extremities and trunk of rowers and kayaker–canoeists and the BMC of the upper extremities in football players lagged behind the age-group average. The BMC of male cyclists and pentathletes was smaller in each region of the body compared to the age-group average. Among female athletes, rhythmic gymnasts and pentathletes had a lower BMC than similar-aged peers in each region of the body (Figure 4, lower part).

The regional development of muscle mass showed a similar pattern (Figure 5). Major differences were apparent in the increased muscular robustness of the upper extremities and the trunk relative to the age-group reference among male wrestlers, rowers, and kayakers/canoeists, and to a lesser extent, among handball players (Figure 5, upper part). On the other hand, the development of muscle mass in the three regions tended to be lower among pentathletes and cyclists and, to a lesser extent, among football players. Among female athletes, rhythmic gymnasts had reduced development of muscle mass in the three regions relative to the age-group reference, while pentathletes had reduced development of muscle mass in the lower extremities compared to the upper extremities and trunk (Figure 5, lower part).

In contrast to BMC and muscle development, the fat mass of the youth athletes showed a similar pattern among the male athletes in different sports (Figure 6, upper part). Cyclists and pentathletes, however, had somewhat less fatness in the different regions compared to similar-aged peers in other sports. Among females, pentathletes and rhythmic gymnasts had considerably less fatness in the three regions than similar-aged peers in handball and basketball (Figure 6, lower part).

The pattern of BMD in the extremities and the trunk expressed as a percentage of total BMD (BMD%) in the total samples of male and female athletes suggested that the mechanical load on the human body varied by the position of the regions relative to posture (bipedal standing or bipedal movement). The higher the position of the region in the human body, the lower the BMD% (Figure 7).

Mean z-scores for the BMD (expressed as a percentage of stature) of the extremities and trunk among athletes in the different sports are illustrated in Figure 8. Overall, the trend indicated asymmetric BMD development by sport. The BMD of the upper extremities and trunk of wrestlers and of the upper extremities of handball players was higher, while the BMD in the lower extremities and trunk of rowers, kayaker–canoeists, and pentathletes was lower relative to the age-group average for male athletes (Figure 8, upper part). Compared to the age-group average, BMD in the three regions was reduced among male cyclists. BMD in the three bodily regions was also reduced among female pentathletes and rhythmic gymnasts (Figure 8, lower part).

A DEXA scan usually includes the BMD of the lumbar spine and proximal femur region; both regions are commonly used to estimate the risk of osteopenia, osteoporosis, and fractures. Z-scores for the BMD of the lumbar spine (Figure 9) were estimated with both the GE Lunar Prodigy DEXA scanner software and the respective BMD reference for the total sample of male and female athletes [20]. The correlation between the z-scores in both sexes was >0.99 (*p* < 0.001), but z-scores based on the athlete reference underestimated those based on the GE Lunar unit by 0.59 z-score units in males and 1.1 z-score units in females. Since the BMD of athletes is usually greater than the 90th percentile of the non-athlete/general population reference [20], the enhanced BMD development in athletes would seem to justify the use of a reference based on athletes, specifically in the context of follow-up examinations after training programs. Nevertheless, the use of a non-athlete reference with athletes is also justified as potential underdevelopment of BMD can be diagnosed with this screening reference.

The median z-score of the BMD of lumbar vertebrae L1 to L4 estimated by the GE Lunar Prodigy DEXA software approached the low BMD category (osteopenia, cut-off value, BMD z-score < −1.0) in male cyclists (BMD z-score = −0.82, see Figure 9). Median z-scores of L1–L4 BMD in athletes in other sports showed a similar pattern to that noted for the total BMD of the youth athletes (Figure 9): pentathletes of both sexes and female rhythmic gymnasts had very low BMD in the lumbar spine region, while male wrestlers and basketball and handball players of both sexes had the highest median BMD z-scores in the sample of athletes considered in the present study (Figure 9).

## 4. Discussion

Higher estimates of BMD and a larger bone mass have been described in athletes in several sports compared to non-athletes and the general population [20,27,28]. The results are generally interpreted in the context of positive benefits of systematic physical activity on bone. However, several studies have shown that not all sports have the same bone-related benefits, especially among those where training starts in childhood and adolescence [29,30,31,32].

Overall, the observations of male athletes highlighted the increased skeletal and muscular robustness of wrestlers and increased muscular development of rowers, kayaker–canoeists, and handball players, but they also highlighted the reduced skeletal and muscular robustness of pentathletes and reduced skeletal development of cyclists in male athletes. The observations were generally consistent with those in studies of male athletes in sports with prevailing speed and strength loadings, e.g., wrestling and rowing [33,34]. However, one study was focused on the proximal femur [33], while the other focused on the mesomorphic component of somatotype estimated from anthropometry [34]. Results for female athletes in the present analysis highlighted the increased skeletomuscular development of basketball and handball players and reduced skeletomuscular development among rhythmic gymnasts and pentathletes. The observations were consistent with those of other studies evaluating body structure among young female athletes. Results from a study of rhythmic gymnasts noted that poor energy balance was associated with a lower lean body mass and reduced skeletomuscular development [35], while other studies noted higher estimates of BMD in a number of skeletal sites among female handball and basketball players [36,37,38].

Observations in the three bodily regions considered in the study indicated reduced BMD among male cyclists, female pentathletes, and rhythmic gymnasts, while the BMD of the lower extremities and trunk was reduced among male rowers, kayaker–canoeists, and pentathletes. On the other hand, BMD was increased in the upper extremities and trunk of wrestlers and in the upper extremities of male handball players. The differences in BMD and variations among bodily regions reflect the frequency patterns of use of the bodily regions in the activities associated with training in the respective sports. The lower the position of the region in the body, the greater the weight loading on the region and, in turn, the higher the BMD in both sexes in the specific sports was observed. Morphofunctional asymmetry has been noted in several studies of youth athletes. For example, skeletomuscular asymmetry was noted between the right and left sides of the body among field hockey players [39] and between the dominant and non-dominant arms of tennis players [40]. Nevertheless, asymmetry in skeletomuscular development among bodily regions in youth athletes participating in different types of sports has not been systematically addressed.

Regular physical activity is an important factor in bone health. Physical activity interventions aimed at augmenting bone mineral were consistent with the preceding observations based on comparisons of active and less active youth. Physical activity interventions, e.g., two or three times per week of moderate-to-high-intensity activities, weight-bearing activities of a longer duration (45–60 min), and/or high-impact activities over a shorter duration (10 min), are associated with enhanced BMC in children and adolescents [41,42,43]. Consensus regarding the type and amount of physical activity required for enhanced development of BMD needs more attention [44,45,46]. Both systematic weight training, which forces the body to work against gravity, and isometric training are associated with increased BMD in athletes due to the greater forces acting on muscle and bone tissues [47].

It is important to study bone mineral characteristics of youth athletes in general and of elite athletes in sports characterized by variations in impact forces. The skeletal development of cyclists, pentathletes, and rhythmic gymnasts should be monitored more frequently since bone development among athletes in these sports tends to lag relative to elite athletes of the same chronological age and relative to population-based reference values. There is also a need to systematically consider the cortical and trabecular architecture of bone among youth athletes in different sports and how it relates to variations in training load.

## 5. Limitations

Given the limited number of female athletes, the analysis of sex differences in total body and regional BMD should be interpreted with caution. The lack of an indicator of maturity status in the younger athletes was also a limitation.

## 6. Conclusions

Results of this study confirmed that youth athletes in sports without systematic weight-bearing tend to have reduced BMD compared to peers of the same chronological age in sports with systematic weight-bearing. Decreased skeletomuscular development and lower bone mineral parameters were noted among male pentathletes and cyclists and among female rhythmic gymnasts and pentathletes. In contrast, increased skeletomuscular development and enhanced bone mineral parameters were noted among male wrestlers, rowers, kayaker–canoeists, and handball players and among female basketball and handball players.

Different loading patterns for specific bodily regions associated with the different sports were reflected in the skeletomuscular development of the respective bodily regions in young athletes. Asymmetry of skeletomuscular development was evident not only from the gradient of weight-bearing but also from the pattern of localization of activities associated with training in specific sports, i.e., the differences in the skeletomuscular development of the upper and lower extremities and the trunk reflected the different types of activities associated with the specific sports considered in the study. Of note, cyclists, pentathletes, and rhythmic gymnasts were at a higher risk for the development of inferior bone structure.

Age-and sex-dependent critical cut-off values for the structural parameters of bone density, architecture, and mass should be specified by the type of sport to facilitate the development of training protocols for elite youth athletes and to reduce the risk of bone stress-related injuries. In addition to bone mineral parameters, indicators of bone geometry indicators should be considered in the screening of stress-related bone injuries since bone geometry is also a determinant of the mechanical resistance of bones [48].

Reference values for indicators of body composition and bone mineral are potentially useful in examinations of athletes in specific sports. The observed differences among athletes in different sport types suggest a need for sport-specific reference values, but such references are not currently available.

## Figures and Tables

**Figure 1 ijerph-19-05083-f001:**
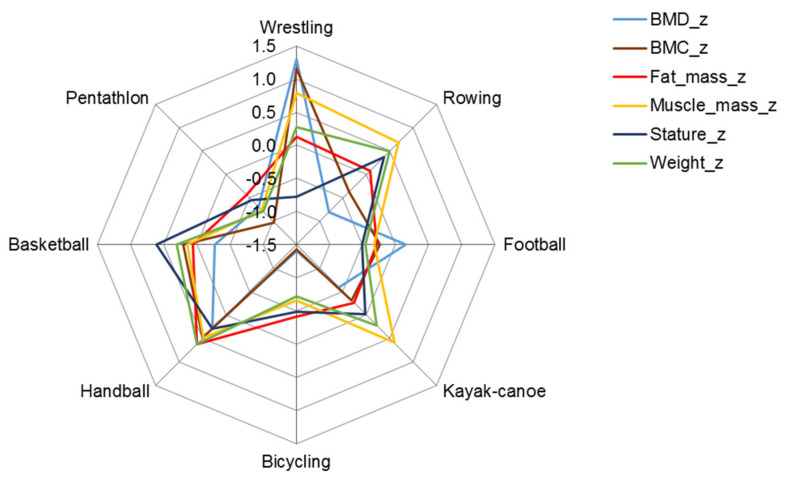
Z profile (mean of z-scores) of structural parameters in male athletes; BMC in grams and fat and muscle mass in kilograms were expressed as a percentage of height in meters.

**Figure 2 ijerph-19-05083-f002:**
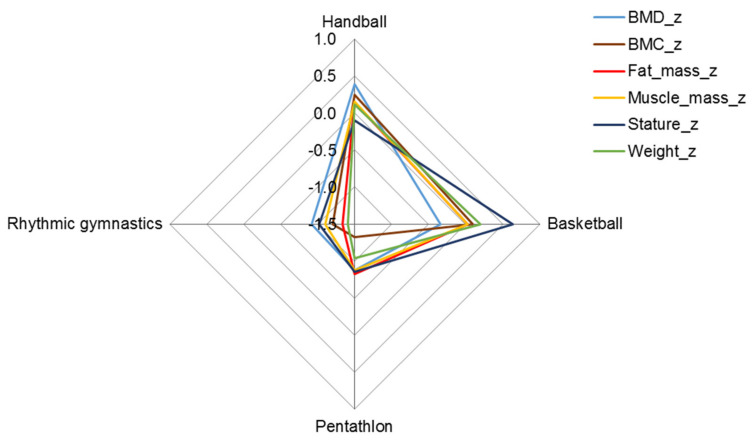
Z profile (mean of z-scores) of structural parameters in female athletes; BMC in grams and fat and muscle mass in kilograms were expressed as a percentage of height in meters.

**Figure 3 ijerph-19-05083-f003:**
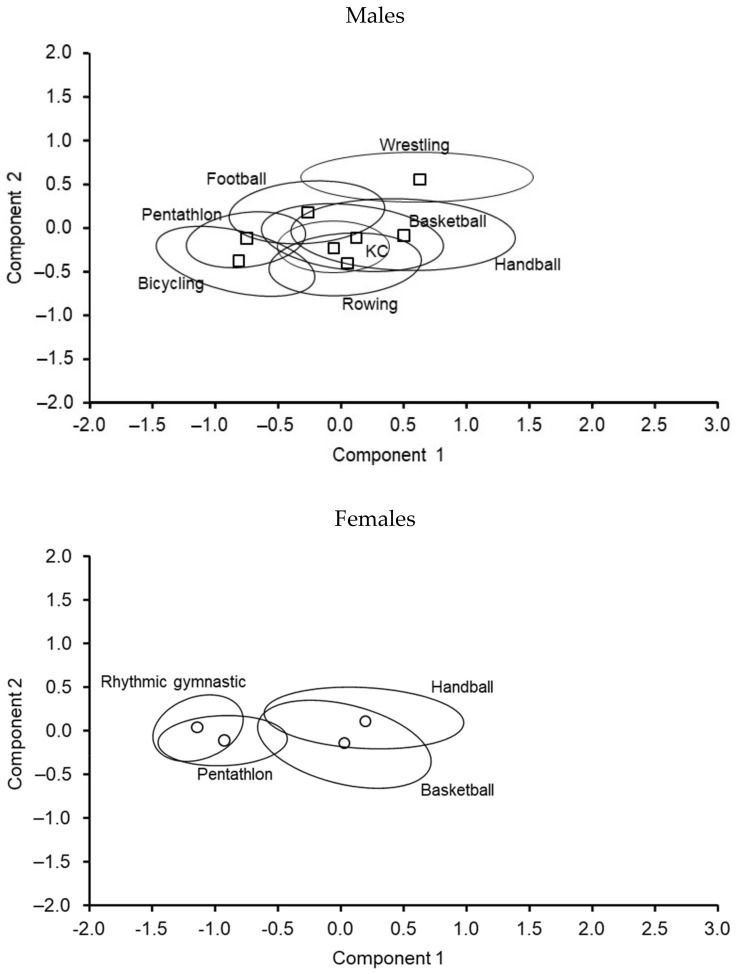
Distribution of PC scores by component (PC 1 on the x-axis and PC 2 on the y-axis) among youth athletes by sport (□ and ○: sport type mean and 95% confidence ellipse around the mean).

**Figure 4 ijerph-19-05083-f004:**
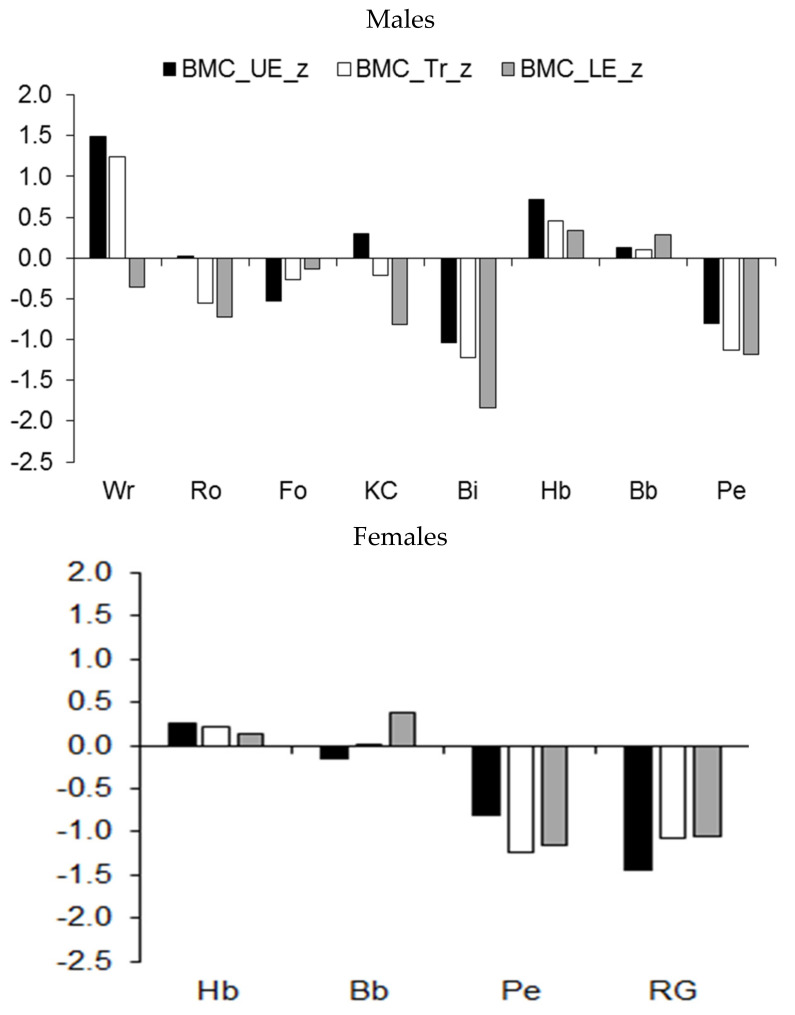
Median z-scores of BMC (expressed as a percentage of stature) of the upper extremities (UE), trunk (Tr), and lower extremities (LE) in youth athletes by sport.

**Figure 5 ijerph-19-05083-f005:**
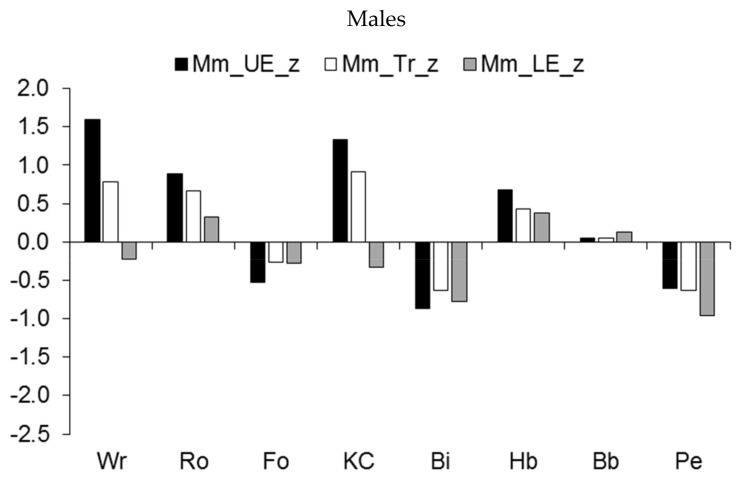
Median z-scores for muscle mass (expressed as a percentage of stature) of the upper extremities (UE), trunk (Tr), and lower extremities (LE) in youth athletes by sport.

**Figure 6 ijerph-19-05083-f006:**
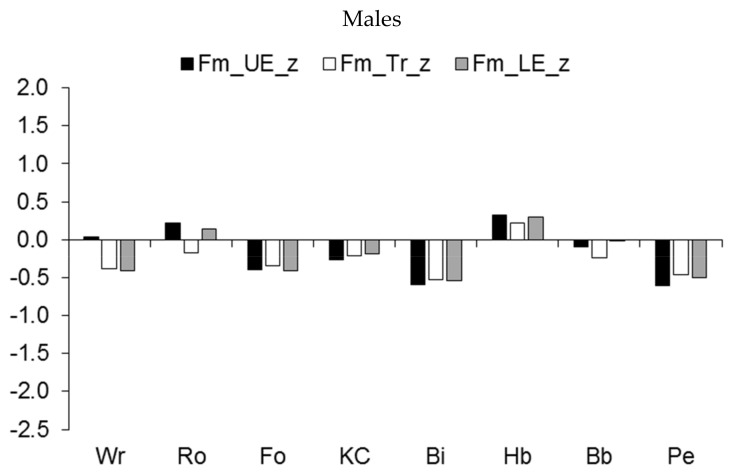
Median z-scores for fat mass (expressed as a percentage of stature) of the upper extremities (UE), trunk (Tr), and lower extremities (LE) in youth athletes by sport.

**Figure 7 ijerph-19-05083-f007:**
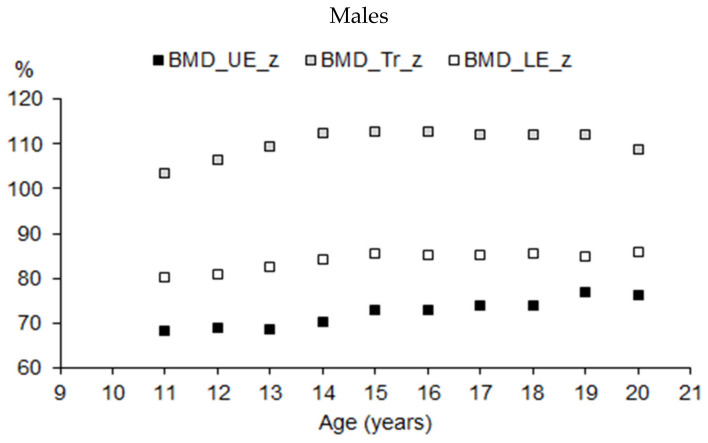
BMD of upper extremities (UE), trunk (Tr), and lower extremities (LE) expressed as a percentage of total BMD by age in youth male and female athletes (Mann–Whitney test: UE—*p* < 0.05 in 13, 14, and 19 years of age; Tr—*p* < 0.05 in 13, 14, and 16 years of age; Le—*p* < 0.05 in the entire interval).

**Figure 8 ijerph-19-05083-f008:**
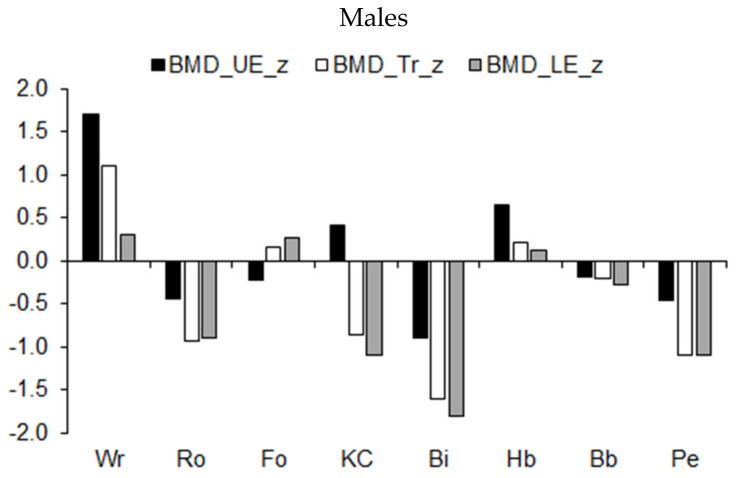
Median z-scores for BMD (expressed as a percentage of stature) of the upper extremities (UE), trunk (Tr), and lower extremities (LE) in youth athletes by sport.

**Figure 9 ijerph-19-05083-f009:**
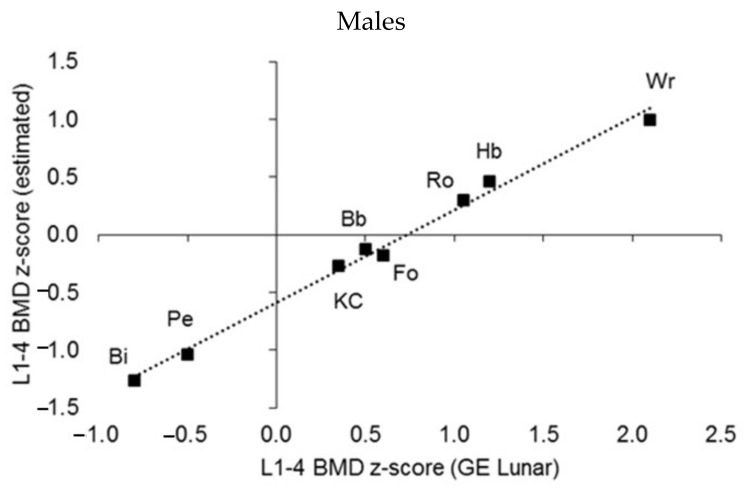
Median L1–L4 BMD z-scores estimated by sport by using the GE Prodigy Lunar reference and the youth athlete reference (Kalabiska et al. 2020); linear regressions: males *p* < 0.001, R2 = 0.991, intercept = −0.59, females *p* < 0.001, R2 = 0.999, intercept = −1.14).

**Table 1 ijerph-19-05083-t001:** Distribution of young athletes by gender within age groups and sports.

Age (Years)	Males	Females	Sports	Males	Females
11	25	-	Wrestling	19	-
12	36	-	Rowing	16	-
13	89	32	Football	531	-
14	138	97	Kayak/canoe	19	-
15	240	103	Bicycling	33	-
16	308	57	Handball	225	251
17	226	71	Basketball	428	139
18	154	31	Pentathlon	28	22
19	50	25	Rhythmic gymnastics	-	23
20	33	19			
Total	1299	435		1299	435

**Table 2 ijerph-19-05083-t002:** Medians for body size, fat and muscle mass, BMC, and BMD in the young male and female athletes by age group.

Age (Years)	Height (cm)	Weight (kg)	Fat Mass (kg)	Muscle Mass (kg)	BMC (g)	BMD (g/cm^2^)
Males						
11	155.8	44.5	3.3	31.6	1689.0	0.962
12	163.2	43.7	3.4	33.0	1878.0	1.007
13	169.8	50.9	3.3	38.1	2123.0	1.055
14	176.4	61.2	3.7	48.4	2573.0	1.159
15	179.3	67.9	4.1	53.4	2901.0	1.246
16	179.8	70.0	4.1	55.8	3082.5	1.298
17	182.6	73.5	4.4	59.3	3230.0	1.349
18	183.2	75.0	4.4	60.3	3272.0	1.367
19	180.6	73.8	4.2	60.5	3377.5	1.378
20	178.0	74.2	4.3	60.3	3455.5	1.443
Females						
13	168.5	54.9	5.2	39.0	2193.5	1.111
14	172.5	63.4	6.7	44.1	2573.0	1.241
15	171.4	62.5	6.6	44.4	2584.0	1.226
16	172.3	64.8	6.6	46.3	2699.0	1.284
17	172.6	67.7	7.0	47.1	2753.0	1.295
18	172.7	65.9	7.2	47.6	2840.0	1.307
19	174.4	68.2	6.1	48.6	2949.5	1.356
20	174.1	71.6	6.4	51.7	2983.0	1.376

**Table 3 ijerph-19-05083-t003:** Significance level of the Wilcoxon signed-rank tests (significant values in italics) of z-scores for components of body mass and for bone parameters in young athletes by sport.

Males	Wr	Ro	Fo	KC	Bi	Hb	Bb	Pe
Weight	0.629	*0.003*	*p < 0.001*	0.184	*p < 0.001*	*p < 0.001*	*p < 0.001*	*p < 0.001*
Height	*0.001*	*0.007*	*p < 0.001*	0.629	*0.002*	*p < 0.001*	*p < 0.001*	*p < 0.001*
Muscle mass	*0.006*	*0.002*	*p < 0.001*	*0.001*	*0.003*	*p < 0.001*	*0.009*	*p < 0.001*
Fat mass	0.444	0.836	*p < 0.001*	*0.049*	*0.001*	*p < 0.001*	0.116	*p < 0.001*
BMC	*p < 0.001*	*p < 0.001*	*p < 0.001*	*p < 0.001*	*p < 0.001*	*p < 0.001*	*p < 0.001*	*p < 0.001*
BMD total	*p < 0.001*	*0.006*	*p < 0.001*	*0.003*	*p < 0.001*	*p < 0.001*	*p < 0.001*	*p < 0.001*
BMD UE	*p < 0.001*	0.379	*p < 0.001*	*0.004*	*0.001*	*p < 0.001*	*p < 0.001*	*p < 0.001*
BMD Tr	*0.004*	*0.002*	*p < 0.001*	*p < 0.001*	*p < 0.001*	*0.049*	*p < 0.001*	*p < 0.001*
BMD Le	*p < 0.001*	*0.003*	*p < 0.001*	*p < 0.001*	*p < 0.001*	*p < 0.001*	*p < 0.001*	*p < 0.001*
**Females**	**Hb**		**Bb**		**Pe**		**RG**	
Weight	0.245		*0.008*		*p < 0.001*		*p < 0.001*	
Height	*0.044*		*p < 0.001*		*p < 0.001*		*p < 0.001*	
Muscle mass	0.075		0.908		*p < 0.001*		*p < 0.001*	
Fat mass	0.069		0.735		*p < 0.001*		*p < 0.001*	
BMC	*p < 0.001*		*p < 0.001*		*p < 0.001*		*p < 0.001*	
BMD total	*p < 0.001*		*p < 0.001*		*p < 0.001*		*p < 0.001*	
BMD UE	*p < 0.001*		*p < 0.001*		*0.017*		*p < 0.001*	
BMD Tr	*p < 0.001*		0.315		*p < 0.001*		*p < 0.001*	
BMD Le	*p < 0.001*		*p < 0.001*		*p < 0.001*		*0.002*	

Wr, wrestling; Ro, rowing; Fo, football; KC, kayak–canoe; Bi, bicycling; Hb, handball; Bb, basketball; Pe, pentathlon; RG, rhythmic gymnastics.

**Table 4 ijerph-19-05083-t004:** Results of the principal components analysis (Cronbach’s alpha: 0.946, bold and italic represent the absolute amount of the component loadings).

	Component 1	Component 2
Eigenvalue	3.172	1.074
% of variance	63.43	19.87
BMD_z	0.742	** *1.693* **
BMC_z	** *1.101* **	0.708
Fat mass_z	0.865	* **−0.980** *
Muscle mass_z	** *1.090* **	−0.272
Weight_z	** *1.141* **	−0.797

**Table 5 ijerph-19-05083-t005:** Significance level in Wilcoxon signed-rank test (significant values in italics) of body mass components in the studied body regions (z-values) in young athletes by the type of sport.

Males	Wr	Ro	Fo	KC	Bi	Hb	Bb	Pe
BMC–UE	*p < 0.001*	0.569	*p < 0.001*	*0.013*	*p < 0.001*	*p < 0.001*	0.070	*p < 0.001*
BMC–Tr	*p < 0.001*	*0.045*	*p < 0.001*	0.107	*p < 0.001*	*p < 0.001*	*0.002*	*p < 0.001*
BMC–LE	0.398	*0.034*	*0.002*	*p < 0.001*	*p < 0.001*	*p < 0.001*	*p < 0.001*	*p < 0.001*
Muscle mass–UE	*p < 0.001*	*0.001*	*p < 0.001*	*p < 0.001*	*0.001*	*p < 0.001*	*0.034*	*0.002*
Muscle mass–Tr	*0.022*	*0.002*	*p < 0.001*	*0.001*	*0.003*	*p < 0.001*	*0.042*	*0.001*
Muscle mass–LE	0.872	*0.023*	*p < 0.001*	0.070	*0.003*	*p < 0.001*	*0.001*	*p < 0.001*
Fat mass–UE	0.354	0.234	*p < 0.001*	0.295	*0.002*	*p < 0.001*	0.112	*0.001*
Fat mass–Tr	0.351	0.877	*p < 0.001*	*0.004*	*0.002*	*p < 0.001*	*p < 0.001*	*p < 0.001*
Fat mass–LE	0.184	0.569	*p < 0.001*	0.334	*0.001*	*p < 0.001*	0.345	*0.010*
**Females**	**Hb**			**Bb**	**Pe**		**RG**	
BMC–UE	*p < 0.001*			*0.027*	*p < 0.001*		*p < 0.001*	
BMC–Tr	*p < 0.001*			*0.683*	*p < 0.001*		*p < 0.001*	
BMC–LE	*0.049*			*p < 0.001*	*p < 0.001*		*p < 0.001*	
Muscle mass–UE	*0.008*			*p < 0.001*	*p < 0.001*		*p < 0.001*	
Muscle mass–Tr	0.115			0.740	*0.008*		*0.001*	
Muscle mass–LE	0.204			*0.027*	*0.026*		*p < 0.001*	
Fat mass–UE	0.073			0.240	*p < 0.001*		*p < 0.001*	
Fat mass–Tr	0.210			0.445	*p < 0.001*		*p < 0.001*	
Fat mass–LE	0.208			0.449	*p < 0.001*		*p < 0.001*	

## Data Availability

Tables with descriptive statistics (means, standard deviations) for age, height, weight, body composition, and bone mineral parameters by age groups for football, handball, and basketball in males and handball and basketball in females and for the total samples for wrestling, rowing, kayak/canoe, bicycling, and pentathlon in males and for pentathlon and rhythmic gymnastics in females are available upon request.

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
