# Peer review of "Sport Activity Load and Skeletomuscular Robustness in Elite Youth Athletes"

_ijerph, 2022, doi:10.3390/ijerph19095083_

Round 1
Reviewer 1 Report
Dear authors,
I reviewed your paper - Sport activity load and skeletal-muscular robustness in elite youth athletes and it strong points are number of subjects, length of the research, analysis of the results, but I have some comments and suggestions in order to be accepted for publishing in IJERPH journal.
Abstract
- Don't use reference.
- Exclude the headings: (1) Objectives, (2) Methods, (3) Results, (4) Conclusion.
Introduction
- Most used information are concerning only for your first objective. Add some new data from the recent studies for the second objective - to compare bone mineral parameters of the upper and lower extremities and of the trunk among young athletes participating in different types of sports.
Materials and Methods
- It is better to organize the content: Study design, Subjects, Data collection.
- There is no information about statistical analysis and how do you used in the research.
- How was sample size determined?
- What about the inclusion and exclusion criteria?
- There are differences between number of the subjects in text (line 84) and table 1.
- In table 2, the columns Fat mass and Muscle mass I think it is about percentage and not kg. For instance first line Boys 11 years old - weight 18,9 kg, fat mass 10,1 kg and muscle mass 31,6 kg. Something is wrong. Please check and offer correct data.
- In table 3 use Legend for abbreviation.
Results
1. Check if your statistics were influenced by the number of participants (different values presented in Material and Methods) and values in Table 2.
Discussion
- Must be extended and needs to reflect what you found.
- Which are the BMD benefits for the subjects according to the sport that they practice?
- There are differences concerning bone mineral parameters of the upper and lower extremities and of the trunk among young athletes participating in different types of sports?
Conclusion
- In this section you write the new information, based only on your original declared objectives. Also, no information for second objective.
Institutional Review Board Statement
1. You need to add some information presented in lines 77-83.
Author Response
The Authors would like to express sincere thanks to the Reviewer for careful reading and suggestion for improvement in the paper.
General comment: I reviewed your paper - Sport activity load and skeletal-muscular robustness in elite youth athletes and it strong points are number of subjects, length of the research, analysis of the results, but I have some comments and suggestions in order to be accepted for publishing in IJERPH journal.
Response: We are grateful for the general comment of the Reviewer, the replies to the suggestions and comments are presented in the order of the Reviewer’s comments.
Comment 1: Abstract - Don't use reference.
Comment 2: Abstract - Exclude the headings: (1) Objectives, (2) Methods, (3) Results, (4) Conclusion.
Response: The Authors thank these comments of the Reviewer. The abstract of the manuscript was corrected by following the Reviewer’s suggestions.
Comment 3: Introduction - Most used information are concerning only for your first objective. Add some new data from the recent studies for the second objective - to compare bone mineral parameters of the upper and lower extremities and of the trunk among young athletes participating in different types of sports.
Response: The Authors thank this question of the Reviewer as well. The manuscript was completed according to the Reviewer’s suggestion as follows:
“Skeleto-muscular development is usually significantly asymmetric in favor of the dominant side or region of the body, although increased level of physical activity may help to prevent incorrect body posture, while asymmetric training loads on skeletal muscles may also enhance incorrect posture (Krzykala and Leszczynski 2015, Zemkova et al. 2019, Lijewski et al. 2021). It is not clear, however, if this developmental asymmetry is also manifest in bone mineral parameters of body segments with different training loads. Therefore, a secondary objective of this study is to compare bone mineral parameters of the upper and lower extremities and of the trunk among young athletes participating in different types of sports.”
cited references:
Krzykala, M., Leszczynski, P. (2015). Asymmetry in body composition in female hockey players. Homo, 66(4), 379-386.
Zemkova, E., Poor, O., Jelen, M. (2019). Between-side differences in trunk rota-tional power in athletes trained in asymmetric sports. Journal of Back and Musculo-skeletal Rehabilitation, 32(4), 529-537.
Lijewski, M., Burdukiewicz, A., Pietraszewska, J., Andrzejewska, J., Stachon, A. (2021). Asymmetry of Muscle Mass Distribution and Grip Strength in Professional Handball Players. International Journal of Environmental Research and Public Health, 18(4), 1913.
Comment 4: Materials and Methods - It is better to organize the content: Study design, Subjects, Data collection.
Response: The Authors thank the comment of the Reviewer, the subheadings were inserted in the Materials and Methods section.
Comment 5: There is no information about statistical analysis and how do you used in the research. How was sample size determined? What about the inclusion and exclusion criteria?
Response: The Authors thank this comment of the Reviewer, as well. The Materials and Methods section was completed with all the information on the statistical analyses used in the study. The corrected Statistical analyses section is the following:
“2.4. Statistical analyses
The body mass components and bone mineral density were expressed relative to height (m) to reduce the influence of body size on the specific parameters. The structural and bone mineral parameters of each athlete were converted to z-scores relative to age- and sex-specific mean and standard deviation values estimated for the total sample ((z = (individual value – age-group mean) / age-group standard deviation). Bone mineral density of the upper extremities, lower extremities and trunk was also expressed as a percentage of total BMD to address asymmetries in BMD by type of sport (Esco et al. 2015).
Wilcoxon signed-rank test was used to compare the z-scores of components of body mass and of bone parameters in the athletes by sport (an alpha of 0.05 was used as the cut-off for significance in all analyses). Linear regression analysis was used to evaluate the relationship between the L1-L4 BMD z-scores estimated by the GE Prodigy Lunar reference and the youth athlete reference series.
Principal components analysis (PCA) was used to reduce the body parameters to a smaller set of components that accounted for most of the variance in the variables considered. Sex-specific PCAs were initially done; as the analyses showed similar components and loadings, the analysis was also done for the total sample. The original variables were log transformed to approximate the normality assumption of PCA. Based on eigen values >1.0; loadings >0.90 were used to identify the variables characteristic of the respective components. Cronbach's alpha was used to estimate the reliability of the analysis. Principal component scores were also calculated for each athlete.”
By considering the population size of elite athletes under 21 years of age, the requested confidence level (95%) and margin error (5%) of statistically accurate sample size determination, more than 380 children should have been enrolled into the study. However, a bigger sample was studied, since the distribution of athletes by sports was not consistent in the associations. Our aim was to recruit at least 15 athletes per sport type by considering the sexes. The subsample sizes of the sport types in the studied sample of young athletes were also determined by the orders of the sports federations, associations and clubs contracted with the Research Center for Sport Physiology, University of Physical Education (Hungary).
The exclusion criteria were the followings: those athletes were excluded from the study who
1) had severe degenerative lesions or fractures/deformations in the measurement area;
2) were unable to reach the correct position and/or remain immobile during measurement,
3/ had a very high or low body mass index (i.e., a BMI that could adversely affect the accuracy of measurement process),
4) were exposed to an enhanced X-ray/CT scan several days prior to the study, or
5) were pregnant.
The manuscript was completed by giving these exclusion criteria in the Material and Methods section.
Comment 6: There are differences between number of the subjects in text (line 84) and table 1.
Response: The Authors thank this comment, the manuscript was corrected as follows:
“Subjects included a cross-sectional sample of 1734 athletes, 1299 males (11-20 years) and 435 females (13-20 years) …”
Comment 7: In table 2, the columns Fat mass and Muscle mass I think it is about percentage and not kg. For instance first line Boys 11 years old - weight 18,9 kg, fat mass 10,1 kg and muscle mass 31,6 kg. Something is wrong. Please check and offer correct data.
Response: Table 2 contained incorrect data on body weight and fat mass, the Authors thank this comment of the Reviewer. Table 2 was corrected as follows:
Age (years) |
Height (cm) |
Weight (kg) |
BMD (g/cm2) |
Fat mass (kg) |
Muscle mass (kg) |
BMC (g) |
Boys |
|
|
|
|
|
|
11 |
155,8 |
44,5 |
0,962 |
3,3 |
31,6 |
1689,0 |
12 |
163,2 |
43,7 |
1,007 |
3,4 |
33,0 |
1878,0 |
13 |
169,8 |
50,9 |
1,055 |
3,3 |
38,1 |
2123,0 |
14 |
176,4 |
61,2 |
1,159 |
3,7 |
48,4 |
2573,0 |
15 |
179,3 |
67,9 |
1,246 |
4,1 |
53,4 |
2901,0 |
16 |
179,8 |
70,0 |
1,298 |
4,1 |
55,8 |
3082,5 |
17 |
182,6 |
73,5 |
1,349 |
4,4 |
59,3 |
3230,0 |
18 |
183,2 |
75,0 |
1,367 |
4,4 |
60,3 |
3272,0 |
19 |
180,6 |
73,8 |
1,378 |
4,2 |
60,5 |
3377,5 |
20 |
178,0 |
74,2 |
1,443 |
4,3 |
60,3 |
3455,5 |
Girls |
|
|
|
|
|
|
13 |
168,5 |
54,9 |
1,111 |
5,2 |
39,0 |
2193,5 |
14 |
172,5 |
63,4 |
1,241 |
6,7 |
44,1 |
2573,0 |
15 |
171,4 |
62,5 |
1,226 |
6,6 |
44,4 |
2584,0 |
16 |
172,3 |
64,8 |
1,284 |
6,6 |
46,3 |
2699,0 |
17 |
172,6 |
67,7 |
1,295 |
7,0 |
47,1 |
2753,0 |
18 |
172,7 |
65,9 |
1,307 |
7,2 |
47,6 |
2840,0 |
19 |
174,4 |
68,2 |
1,356 |
6,1 |
48,6 |
2949,5 |
20 |
174,1 |
71,6 |
1,376 |
6,4 |
51,7 |
2983,0 |
Comment 8: In table 3 use Legend for abbreviation.
Response: The manuscript was completed with the abbreviation of Table 3 as follows:
“Wr: wrestling, Ro: rowing, Fo: football, KC: kayak-canoe, Bi: bicycling, Hb: handball, Bb: basketball, Pe: pentathlon, RG: rhythmic gymnastics”
Comment 9: Results - Check if your statistics were influenced by the number of participants (different values presented in Material and Methods) and values in Table 2.
Response: The Authors thank this comment of the Reviewer. The statistics were correct in the manuscript, the data in Table 2 were also correct, only the misspelling in the beginning of the Material and Methods were corrected.
Comment 10: Discussion - Must be extended and needs to reflect what you found. Which are the BMD benefits for the subjects according to the sport that they practice? There are differences concerning bone mineral parameters of the upper and lower extremities and of the trunk among young athletes participating in different types of sports?
Response: The Authors thank these comments. The Discussion section was completed with the following paragraphs on the skeleton-muscular and BMD results:
“Overall, the observations for male athletes highlight the increased skeletal and muscular robustness of wrestlers and increased muscular development of rowers, kayaker-canoeists and handball players, but also highlight the decreased skeletal and muscular robustness of pentathletes and reduced skeletal development of cyclists in male athletes. Among female athletes, the results highlight the increased skeletal-muscular development of basketball and handball players, but reduced skeletal-muscular development among rhythmic gymnasts and pentathletes.
Observations in the three bodily regions considered in the study indicated reduced BMD among male cyclist, female pentathletes and rhythmic gymnasts, while BMD of the lower extremities and trunk was reduced among male rowers, kayaker-canoeists and pentathletes. On the other hand, BMD was increased in the upper extremities and trunk of wrestlers and in the upper extremities of male handball players. The differences in BMD and variation among bodily regions reflect the frequency pattern of use of the bodily regions in the activities associated with training in the respective sports. The lower the position of the region in the body, the greater the weight loading on the region and in turn a higher BMD in both sexes in the specific sports considered.”
Comment 11: Conclusion - In this section you write the new information, based only on your original declared objectives. Also, no information for second objective.
Response: This comment was very helpful in the correction of the manuscript, the Conclusion section was completed as follows:
“Results of the present study confirmed that youth athletes in sports without systematic weight-bearing tend to have reduced BMD compared to peers of the same chronological age in sports with systematic weight-bearing. Decreased skeleto-muscular development and lower bone mineral parameters were noted among male pentathletes and cyclists, and among female rhythmic gymnasts and pentathletes. In contrast, the increased skeleton-muscular development and enhanced bone mineral parameters were noted among male wrestlers, rowers, kayaker-canoeists and handball players, and among female basketball and handball players.
The different loading pattern for specific bodily regions associated with the different sport types were reflected in the skeleton-muscular development of the respective bodily regions in young athletes. Asymmetry of skeleton-muscular development was evident not only by the gradient of weight loading, but also by the pattern of localization of activities associated with training in specific sports, i.e., the differences in the skeleton-muscular development of the upper and lower extremities and the trunk reflected the differences types of sport activity associated with the specific sports considered in the study. Of note, cyclists, pentathletes and rhythmic gymnasts were at higher risk for the development of inferior bone structure.”
Comment 12: Institutional Review Board Statement - You need to add some information presented in lines 77-83.
Response: The Research Center for Sport Physiology, University of Physical Education (Hungary) has a cooperation network with numerous sports federations, associations and clubs in the studied sport types.”, however the contracts with these sports bodies are not open to public to GDPR restrictions. The statement was completed with the following information:
“The project was approved by the Research Ethics Committee of the University of Physical Education in Budapest, Hungary (ID of approval: TE-KEB/No42/2019). The Research Center for Sport Physiology (University of Physical Education, Hungary) has a cooperation agreement with numerous sports federations, associations and clubs that focus on a variety of sports. The governing bodies of the respective sport organizations also approved the ethical codes established by Research Ethics Committee of the University. The research was carried out following the rules of the Declaration of Helsinki of 1975 (as revised in 2013. Parents of athletes <18 years and also the athletes were informed of the details of the project; both provided written informed consent. Details of the project were also provided for older athletes who also provided informed written consent.”
Reviewer 2 Report
The main aim of the paper „Sport Activity Load and Skeletal-Muscular Robustness in Elite Youth Athletes“ is to compare variation in bone mineral parameters of the young athletes participating in sports with different mechanical loads.
The topic is interesting. I would like to congratulate the authors for an interesting study. There are just a few details that need to be explained.
Line 33: Based on the paper aim, I recommend to also mention the connection with youth athletes. E.g. with the sensitive period of bone accrual and the cruciality of the bone development in children and youth. For all e.g.: Stagi, S., Cavalli, L., Iurato, C., Seminara, S., Brandi, M. L., & de Martino, M. (2013). Bone metabolism in children and adolescents: main characteristics of the determinants of peak bone mass. Clinical cases in mineral and bone metabolism : the official journal of the Italian Society of Osteoporosis, Mineral Metabolism, and Skeletal Diseases, 10(3), 172–179.
Line 76: I recommend to divide chapter 2 Materials and Methods into other sub-chapters, for better clarity of the text.
Line 84: I recommend to edit the text, e.g.: Subjects were 1793 athletes, 1316 males (11-20 years old) and 477 females (13-20 years old), who volunteered to participate in this cross-sectional study (Tables 1-2) [17].
Line 92: From what the day time, when were the measurements taken? Were subject somehow instructed about food intake or drink consuption prior to the measurements? What about training regime prior to the measurements? Based on e.g. Toomey, McCormack, & Jakeman (2017) I would highly recommend to add those information. The effect of hydration status on the measurement of lean tissue mass by dual-energy X-ray absorptiometry. European journal of applied physiology, 117(3), 567–574. https://doi.org/10.1007/s00421-017-3552-x
Line 94: I recommend to put in brackets after the name of the device: manufacturer, city, country of origin of the device.
Line 99: Add a description of segmental determination. Use a citation or briefly description.
Line 109: Mostly used is Total instead of together
Line 110: Why is the median used instead of the mean and standard deviation? For clarity I recommend to put in columns next to each other: BMC and BMD.
Line 132: This is the first time to use those abbrevation I would reccomend to describe them prior to this table. E.g. use it in the Table 1 as Wrestling (Wr)
Line 142: The highlighting looks odd. The font (the size) does not fit with the rest of the article as well. I recommend to edit.
Line 264: I miss the comparison with the results from this paper.
Line 288: Knowing that this is not main aim of the article: In the paper is mention the risk of bone stress-related injuries. Due to this I can see another limitation. From DXA there is information only about BMD and BMC without the information about bone gometry, which can play also a role in case of bone fracture. Bone and geometry: Szulc P. (2006). Bone density, geometry, and fracture in elderly men. Current osteoporosis reports, 4(2), 57–63. https://doi.org/10.1007/s11914-006-0003-8; Fonseca, H., Moreira-Gonçalves, D., Coriolano, H. J., & Duarte, J. A. (2014). Bone quality: the determinants of bone strength and fragility. Sports medicine (Auckland, N.Z.), 44(1), 37–53. https://doi.org/10.1007/s40279-013-0100-7
Line 292: I recommend to add short comparison with the reference values for sports? The whole paper is focused on the comparison with reference vaules. Other conclusions should be make with high caution.
Line 338: References are incorrectly numbered.
Author Response
The Authors would like to express sincere thanks to the Reviewer for careful reading and suggestion for improvement in the paper.
General comment: The main aim of the paper „Sport Activity Load and Skeletal-Muscular Robustness in Elite Youth Athletes“ is to compare variation in bone mineral parameters of the young athletes participating in sports with different mechanical loads.
The topic is interesting. I would like to congratulate the authors for an interesting study. There are just a few details that need to be explained.
Response: We are grateful for the general comment of the Reviewer, the replies to the suggestions and comments are presented in the order of the Reviewer’s comments.
Comment 1: Line 33: Based on the paper aim, I recommend to also mention the connection with youth athletes. E.g. with the sensitive period of bone accrual and the cruciality of the bone development in children and youth. For all e.g.: Stagi, S., Cavalli, L., Iurato, C., Seminara, S., Brandi, M. L., & de Martino, M. (2013). Bone metabolism in children and adolescents: main characteristics of the determinants of peak bone mass. Clinical cases in mineral and bone metabolism: the official journal of the Italian Society of Osteoporosis, Mineral Metabolism, and Skeletal Diseases, 10(3), 172–179.
Response: The Authors thank this comment of the Reviewer, the manuscript was completed in the first paragraph of Introduction as follows:
“Athletes are not a homogenous population as both intrinsic (genetic, biological) and extrinsic (environmental, nutritional, training) factors affect performance [1]. Both regular training for sport [2] and regular participation in physical activity [3] generally have a beneficial effect on mineral accrual and thus bone mass and in turn bone mineral density (BMD). The skeleton in childhood and adolescence is sensitive to the mechanical stimulation elicited by physical activity so that regular physical activity during childhood and adolescence can optimize skeletal health that persists through adulthood. Factors other than mechanical stimulation also influence bone development and include genotype and adequate levels of vitamin D and calcium (Vicente-Rodriguez 2006, Moreno et al. 2011, Stagi et al. 2013).”
cited references:
Vicente-Rodriguez, G. (2006). How does exercise affect bone development during growth? Sports Medicine, 36(7), 561-569.
Stagi, S., Cavalli, L., Iurato, C., Seminara, S., Brandi, M. L., & de Martino, M. (2013). Bone metabolism in children and adolescents: main characteristics of the determinants of peak bone mass. Clinical Cases in Mineral and Bone Metabolism, 10(3), 172.
Comment 2: Line 76: I recommend to divide chapter 2 Materials and Methods into other sub-chapters, for better clarity of the text.
Response: The Authors thank the comment of the Reviewer, subheadings were inserted in the Materials and Methods section
Comment 3: Line 84: I recommend to edit the text, e.g.: Subjects were 1793 athletes, 1316 males (11-20 years old) and 477 females (13-20 years old), who volunteered to participate in this cross-sectional study (Tables 1-2) [17].
Response: The Authors thank this comment, the manuscript was corrected according to the Reviewer’s suggestion.
Comment 4: Line 92: From what the day time, when were the measurements taken? Were subject somehow instructed about food intake or drink consumption prior to the measurements? What about training regime prior to the measurements? Based on e.g. Toomey, McCormack, & Jakeman (2017) I would highly recommend to add those information. The effect of hydration status on the measurement of lean tissue mass by dual-energy X-ray absorptiometry. European journal of applied physiology, 117(3), 567–574. https://doi.org/10.1007/s00421-017-3552-x
Response: The Authors thank this comment, as well. The manuscript was completed as follows:
“Subjects were asked to avoid eating and drinking for at least 60 minutes prior to examination, and also to follow their habitual training regime during the week of the examination. All examinations took place between 9.00 and 12.00 in the morning.”
Comment 5: Line 94: I recommend to put in brackets after the name of the device: manufacturer, city, country of origin of the device.
Response: The Authors thank the comment, the manuscript was corrected according to the Reviewer’s suggestion.
Comment 6: Line 99: Add a description of segmental determination. Use a citation or briefly description.
Response: The Authors thank this comment, the manuscript was corrected. The following citation was added:
Esco, M. R., Snarr, R. L., Leatherwood, M. D., Chamberlain, N. A., Redding, M. L., Flatt, A. A., Williford, H. N. (2015). Comparison of total and segmental body composition using DXA and multifrequency bioimpedance in collegiate female athletes. The Journal of Strength & Conditioning Research, 29(4), 918-925.
Comment 7: Line 109: Mostly used is Total instead of together
Response: The Authors thank this comment, the manuscript was corrected.
Comment 8: Line 110: Why is the median used instead of the mean and standard deviation? For clarity I recommend to put in columns next to each other: BMC and BMD.
Response: Table 2 was corrected according the Reviewer’s suggestion. Median values are used as a possible and frequently used measure of central tendency. Further statistical data can be required as it is mentioned in Data Availability Statement: “Supplementary tables with descriptive statistics (means, standard deviations) for age, height, weight, body composition and bone mineral parameters by age groups for football, handball and basketball in males and handball and basketball in females) and for the total samples for wrestling, rowing, kayak-canoe, bicycling and pentathlon in males and for pentathlon and rhythmic gymnastics in females are available upon request.”
Comment 9: Line 132: This is the first time to use those abbreviation I would recommend to describe them prior to this table. E.g. use it in the Table 1 as Wrestling (Wr)
Response: Table 1 was corrected according to the Reviewer’s suggestion, the Authors are thankful for this comment, as well.
Comment 10: Line 142: The highlighting looks odd. The font (the size) does not fit with the rest of the article as well. I recommend to edit.
Response: The Authors thank this comment, Table 3 was corrected according to the Reviewer’s suggestion.
Comment 11: Line 264: I miss the comparison with the results from this paper.
Response: The Author thank this comment of the Reviewer. The Discussion section was completed with the following paragraphs:
“Overall, the observations for male athletes highlight the increased skeletal and muscular robustness of wrestlers and increased muscular development of rowers, kayaker-canoeists and handball players, but also highlight the decreased skeletal and muscular robustness of pentathletes and reduced skeletal development of cyclists in male athletes. Among female athletes, the results highlight the increased skeletal-muscular development of basketball and handball players, but reduced skeletal-muscular development among rhythmic gymnasts and pentathletes.
Observations in the three bodily regions considered in the study indicated reduced BMD among male cyclist, female pentathletes and rhythmic gymnasts, while BMD of the lower extremities and trunk was reduced among male rowers, kayaker-canoeists and pentathletes. On the other hand, BMD was increased in the upper extremities and trunk of wrestlers and in the upper extremities of male handball players. The differences in BMD and variation among bodily regions reflect the frequency pattern of use of the bodily regions in the activities associated with training in the respective sports. The lower the position of the region in the body, the greater the weight loading on the region and in turn a higher BMD in both sexes in the specific sports considered.”
Comment 12: Line 288: Knowing that this is not main aim of the article: In the paper is mention the risk of bone stress-related injuries. Due to this I can see another limitation. From DXA there is information only about BMD and BMC without the information about bone geometry, which can play also a role in case of bone fracture. Bone and geometry: Szulc P. (2006). Bone density, geometry, and fracture in elderly men. Current osteoporosis reports, 4(2), 57–63. https://doi.org/10.1007/s11914-006-0003-8; Fonseca, H., Moreira-Gonçalves, D., Coriolano, H. J., & Duarte, J. A. (2014). Bone quality: the determinants of bone strength and fragility. Sports medicine (Auckland, N.Z.), 44(1), 37–53. https://doi.org/10.1007/s40279-013-0100-7
Response: The Authors agree this comment, the Conclusion section was completed with the following, closing sentence:
“In addition to bone mineral parameters, indicators of bone geometry indicators should be considered in the screening of stress-related bone injuries, since bone geometry is also a determinant of the mechanical resistance of bones (Szulc 2006).”
the cited reference:
Szulc, P. (2006). Bone density, geometry, and fracture in elderly men. Current Osteoporosis Reports, 4(2), 57-63.
Comment 13: Line 292: I recommend to add short comparison with the reference values for sports? The whole paper is focused on the comparison with reference values. Other conclusions should be make with high caution.
Response: Reference values for sports are missing from the literature of bone mineral variables, sometimes references values for athletes are also missing, but references for types of sports in elite athletes are absolutely missing actually. That is why the mentioned part of the manuscript does not contain a comparison with the reference values for sports. The manuscript was completed with a sentence in the Conclusion section to mention this problem as follows:
“Reference values for indicators of body composition and bone mineral are potentially useful in examinations of athletes in specific sports. The observed differences among athletes in different sport types would seemingly suggest a need for sport-specific reference values, but such references are not presently available.”
Comment 14: Line 338: References are incorrectly numbered.
Response: References were corrected.
Reviewer 3 Report
This manuscript is interesting; several points require clarification. The type of research design should be explicitly established, inferring from the materials section ¿it should be a cross-sectional multi-secular study?.
Also, other variables should be mentioned if they were recorded or if the study lack this information about the hormonal development status in those boys and girls growth population.
Finally, the included sample by each sport represents a directed selection of athletes? Are they representing the top performers of their origin cohorts? Alternatively, where are they randomly selected?.
Author Response
The Authors would like to express sincere thanks to the Reviewer for careful reading and suggestion for improvement in the paper.
General comment: This manuscript is interesting; several points require clarification.
Response: The replies to the suggestions and comments are presented in the order of the Reviewer’s comments.
Comment 1: The type of research design should be explicitly established, inferring from the materials section - it should be a cross-sectional multi-secular study?
Response: The Authors thank this comment of the Reviewer. The study was a cross-sectional study, conducted between 2015 and 2020. Each athlete participated in the examination only once. Since the examinations were carried out only between 9.00 and 12.00 in the mornings, and a DEXA examination takes a longer time compared to general anthropometric examinations, this 5-year long interval required to collect data on athletes’ body and bone structural measurements, which database was big enough to analyses body and bone structural differences among sport types.
The manuscript was completed a follows:
“The cross-sectional research was conducted between September 2015 and February 2020.”
Comment 2: Also, other variables should be mentioned if they were recorded or if the study lack this information about the hormonal development status in those boys and girls growth population.
Response: The Authors thank the comment of the Reviewer. Only anthropometric and biomechanical examinations happened before the DEXA examinations, not other endocrinological examinations were carried out in the studies sample of young athletes.
Comment 3: Finally, the included sample by each sport represents a directed selection of athletes? Are they representing the top performers of their origin cohorts? Alternatively, where are they randomly selected?
Response: The sample represent all the athletes of the sports federations, associations and clubs having examination contracts with Research Center for Sport Physiology, University of Physical Education (Hungary), not only the top performers of their origin cohorts in their federations, associations and clubs. Yes, the were randomly selected from the athletes of federations, associations and clubs from their age-groups.
Reviewer 4 Report
The paper is interesting and in general well written however should be proofreaded by native speaker (Authors can ask co-author profesor dr h.c Malina to proofread text - for example line 84 Subjects were 1793 were athletes,…)
I have some minor comments mainly concerning adding some additional references in terms of citing relevant works
Introduction
- In the Intro I recommend to add one-two sentences about preventing role of witamin D in context of bone growth
I suggest to citate below atrticle
http://tss.awf.poznan.pl/files/2016/vol%2023%20no%202/2_Dalz_TSS_2016_22363-71.pdf
or other article in this subject
- It would be worthy to mention that PA is in general related with prevention against incorrect body posturÄ™ among young people however some sports may enhance incorrect body posture
Please you can citate paper below
Material and methods
This part is correct and well written
Discussion
In the second paragraph I suggest to refer to some papers that amphasized there is not 100% consensus what weekly volume and intensity should be recommended since we really do not know what is here real „pattern from Sevres near Paris”.
References
Please correct numberng and than re-number citations in main text – you have twice #1
- Yingling, V.R.; Ferrari-Church, B.; Strickland, A. Tibia functionality and Division II female and male collegiate athletes from 338
multiple sports. Peer J. 2018, 6: e5550 doi: https://doi.org/10.7717/peerj.5550 339
1. Andreoli, A.M.; Monteleone, M.; Van Loan, L.; Promenzio, U.; Tarantino, A.; De Lorenzo. Effects of different sports on bone 340
density and muscle mass in highly trained athletes. Med. Sci. Sports Exerc. 2001, 33(4): 507–511. doi: 341
https://doi.org/10.1097/00005768-200104000-00001 342
2. Nelson, M.E.; Rejeski, W.J.; Blair, S.N.; Duncan, P.W.; Judge, J.O.; King, A.C.; Macera, C.A.; Castaneda-Sceppa, C. Physical ac- 343
tivity and public health in older adults: recommendation from the American College of Sports Medicine and the
Author Response
The Authors would like to express sincere thanks to the Reviewer for careful reading and suggestion for improvement in the paper.
General comment: The paper is interesting and in general well written. I have some minor comments mainly concerning adding some additional references in terms of citing relevant works.
Response: The replies to the suggestions and comments are presented in the order of the Reviewer’s comments.
Comment 1: Introduction - In the Intro I recommend to add one-two sentences about preventing role of vitamin D in context of bone growth. I suggest to citate below atrticle: http://tss.awf.poznan.pl/files/2016/vol%2023%20no%202/2_Dalz_TSS_2016_22363-71.pdf
or other article in this subject
Response: The Authors thank this comment of the Reviewer. The Introduction section was completed according to the Reviewer’s suggestion as follows:
“The skeleton in childhood and adolescence is sensitive to the mechanical stimulation elicited by physical activity so that regular physical activity during childhood and adolescence can optimize skeletal health that persists through adulthood. Factors other than mechanical stimulation also influence bone development and include genotype and adequate levels of vitamin D and calcium (Vicente-Rodriguez 2006, Moreno et al. 2011, Stagi et al. 2013).”
cited reference:
Moreno, L. A., Valtuena, J., Pérez-López, F., & González-Gross, M. (2011). Health effects related to low vitamin D concentrations: beyond bone metabolism. Annals of Nutrition and Metabolism, 59(1), 22-27.
Comment 2: It would be worthy to mention that PA is in general related with prevention against incorrect body posturÄ™ among young people however some sports may enhance incorrect body posture. Please you can citate paper below:
Response: The Authors thank this comment, as well. The Introduction section was completed according to the Reviewer’s suggestion as follows:
“The asymmetry of skeleto-muscular development between the upper and lower body segments (e.g. rowers versus cyclers), or between the dominant and non-dominant arms (e.g. tennis players versus swimmers) associated with training load is reasonably well established. Skeleto-muscular development is usually significantly asymmetric in favor of the dominant side or region of the body, although increased level of physical activity may help to prevent incorrect body posture, while asymmetric training loads on skeletal muscles may also enhance incorrect posture (Krzykala and Leszczynski 2015, Zem-kova et al. 2019, Lijewski et al. 2021, Veis et al. 2022).”
cited reference:
Veis, A., Kanásová, J., Halmová, N. (2022) The level of body posture, the flexibility of backbone and flat feet in competition fitness in 8-11year old girls. TRENDS in Sport Sciences 2022; 29(1): 5-11.
Comment 3: Discussion - In the second paragraph I suggest to refer to some papers that amphasized there is not 100% consensus what weekly volume and intensity should be recommended since we really do not know what is here real „pattern from Sevres near Paris”.
Response: The Discussion section was completed according to the Reviewer’s suggestion as follows:
“Consensus as to the type and amount of physical activity required for enhanced de-velopment of BMD needs more attention (Nikander et al. 2010, O’Donovan et al. 2010, Graf et al. 2014).”
cited references
O'Donovan, G., Blazevich, A. J., Boreham, C., Cooper, A. R., Crank, H., Ekelund, U., Stamatakis, E. (2010). The ABC of Physical Activity for Health: a consensus statement from the British Association of Sport and Exercise Sciences. Journal of sports sciences, 28(6), 573-591.
Graf, C., Beneke, R., Bloch, W., Bucksch, J., Dordel, S., Eiser, S., Woll, A. (2014). Recommendations for promoting physical activity for children and adolescents in Germany. A consensus statement. Obesity facts, 7(3), 178-190.
Nikander, R., Sievänen, H., Heinonen, A., Daly, R. M., Uusi-Rasi, K., Kannus, P. (2010). Targeted exercise against osteoporosis: a systematic review and meta-analysis for optimising bone strength throughout life. BMC medicine, 8(1), 1-16.
Comment 4: References - Please correct numbern and than re-number citations in main text – you have twice #1
Response: References were corrected.
Round 2
Reviewer 1 Report
You made changes according to my suggestions and comments. In my opinion the paper is suitable for publication.
Author Response
The Authors would like to express sincere thanks to the Reviewer for careful reading and suggestions for improvement in the paper.
Reviewer 2 Report
I thank the authors for explaining and adding to the text. I have only one comment: the results achieved are already listed and commented on in the discussion, but I recommend comparing them better with already published studies.
Author Response
The Authors would like to express sincere thanks to the Reviewer for careful reading and suggestion for improvement in the paper.
General comment: I thank the authors for explaining and adding to the text. I have only one comment: the results achieved are already listed and commented on in the discussion, but I recommend comparing them better with already published studies.
Response: The comment of the Reviewer is accepted, the manuscript was completed in the Discussion section by comparing the presented results with formerly published studies as follows:
Overall, the observations for male athletes highlighted the increased skeletal and muscular robustness of wrestlers and increased muscular development of rowers, kayaker-canoeists and handball players, but also highlight the reduced skeletal and muscular robustness of pentathletes and reduced skeletal development of cyclists in male athletes. The observations were generally consistent with those in studies of male athletes in sports with prevailing speed and strength loadings, e.g., wrestling and rowing (Bozkurt 2010; Kutseryb et al 2017). Note, however, one study was focused on the proximal femur (Boskurt, 2010), while the other focused on the mesomorphic component of somatotype estimated from anthropometry (Kutseryb et al., 2017). Results for female athletes in the present analysis highlighted the increased skeletal-muscular development of basketball and handball players, and the reduced skeletal-muscular development among rhythmic gymnasts and pentathletes. The observations were consistent with those of other studies evaluating body structure among young female athletes. Results from a study of rhythmic gymnasts noted that poor energy balance was associated with a lower lean body mass and reduced skeletal-muscular development (Blake 2015), while other studies noted higher estimates of BMD in a number of skeletal sites among female handball and basketball players (Moss et al. 2015, Pastuszak et al. 2018, Stojanovic et al 2020).
Observations in the three bodily regions considered in the study indicated reduced BMD among male cyclist, female pentathletes and rhythmic gymnasts, while BMD of the lower extremities and trunk was reduced among male rowers, kayaker-canoeists and pentathletes. On the other hand, BMD was increased in the upper extremities and trunk of wrestlers and in the upper extremities of male handball players. The differences in BMD and variation among bodily regions reflect the frequency pattern of use of the bodily regions in the activities associated with training in the respective sports. The lower the position of the region in the body, the greater the weight loading on the region and in turn a higher BMD in both sexes in the specific sports considered. Observations in the three bodily regions considered in the study indicated reduced BMD among male cyclist, female pentathletes and rhythmic gymnasts, while BMD of the lower extremities and trunk was reduced among male rowers, kayaker-canoeists and pentathletes. On the other hand, BMD was increased in the upper extremities and trunk of wrestlers and in the upper extremities of male handball players. The differences in BMD and variation among bodily regions reflect the frequency pattern of use of the bodily regions in the activities associated with training in the respective sports. The lower the position of the region in the body, the greater the weight loading on the region and in turn a higher BMD in both sexes in the specific sports considered. Morphofunctional asymmetry has been noted in several studies of youth athletes. For example, skeletal-muscular asymmetry was noted between the right and left sides of the body among field hockey players (Krzykała et al. 2018) and between the dominant and non-dominant arms of tennis players (Sanchis-Moysi et al. 2010). Nevertheless, asymmetry in skeletal-muscular development among bodily regions in youth athletes participating in different types of sport has not been systematically addressed.
cited references:
Bozkurt, I. (2010). Effects of exercises on bone mineral density of proximal femour region among athletes of different branches. International Journal of Physical Sciences, 5(17), 2705-2714.
Kutseryb, T., Vovkanych, L., Hrynkiv, M., Majevska, S., Muzyka, F. (2017). Peculiarities of the somatotype of athletes with different directions of the training process. Journal of Physical Education and Sport, 17(1), 431.
Blake, T. E. (2015). Relationship of Energy Balance and Body Composition in Elite Female Gymnasts.
Moss SL, McWhannell N, Michalsik LB, Twist C. 2015. Anthropometric and physical per-formance characteristics of top-elite, elite and non-elite youth female team handball players. J Sports Sci 33(17):1780-89
Pastuszak, A., Górski, M., Gajewski, J., BuÅ›ko, K. (2018). Anthropometric profile of female handball players is related to bone mineral density. AnthropologicAl review, 81(3), 298-306.
Stojanović, E., Radovanović, D., Dalbo, V. J., Jakovljević, V., Ponorac, N., Agostinete, R. R., Scanlan, A. T. (2020). Basketball players possess a higher bone mineral density than matched non-athletes, swimming, soccer, and volleyball athletes: a systematic review and meta-analysis. Archives of osteoporosis, 15(1), 1-21.
KrzykaÅ‚a, M., LeszczyÅ„ski, P., GrzeÅ›kowiak, M., Podgórski, T., Woźniewicz-DobrzyÅ„ska, M., Konarska, A., Konarski, J. M. (2018). Does field hockey increase morphofunctional asymmetry? A pilot study. Homo, 69(1-2), 43-49.
Sanchis-Moysi, J., Dorado, C., Olmedillas, H., Serrano-Sanchez, J. A., Calbet, J. A. (2010). Bone and lean mass inter-arm asymmetries in young male tennis players depend on training frequency. European journal of applied physiology, 110(1), 83-90.